# DFT Investigation of Hydrogen Atom Abstraction from NHC-Boranes by Methyl, Ethyl and Cyanomethyl Radicals—Composition and Correlation Analysis of Kinetic Barriers

**DOI:** 10.3390/molecules25194509

**Published:** 2020-10-01

**Authors:** Hong-jie Qu, Lang Yuan, Cai-xin Jia, Hai-tao Yu, Hui Xu

**Affiliations:** 1Key Laboratory of Functional Inorganic Material Chemistry (Ministry of Education) and School of Chemistry and Materials Science, Heilongjiang University, Harbin 150080, China; qhjsxm@163.com (H.-j.Q.); 2020007@hlju.edu.cn (L.Y.); jiacx@hlju.edu.cn (C.-x.J.); 2College of Science, Heilongjiang Bayi Agricultural University, Daqing 163319, China

**Keywords:** NHC-boranes, hydrogen atom abstraction, correlation analysis, kinetic barrier, bond dissociation energy

## Abstract

Understanding the hydrogen atom abstraction (HAA) reactions of *N*-heterocyclic carbene (NHC)-boranes is essential for extending the practical applications of boron chemistry. In this study, density functional theory (DFT) computations were performed for the HAA reactions of a series of NHC-boranes attacked by ^•^CH_2_CN, Me^•^ and Et^•^ radicals. Using the computed data, we investigated the correlations of the activation and free energy barriers with their components, including the intrinsic barrier, the thermal contribution of the thermodynamic reaction energy to the kinetic barriers, the activation Gibbs free energy correction and the activation zero-point vibrational energy correction. Furthermore, to describe the dependence of the activation and free energy barriers on the thermodynamic reaction energy or reaction Gibbs free energy, we used a three-variable linear model, which was demonstrated to be more precise than the two-variable Evans–Polanyi linear free energy model and more succinct than the three-variable Marcus-theory-based nonlinear HAA model. The present work provides not only a more thorough understanding of the compositions of the barriers to the HAA reactions of NHC-boranes and the HAA reactivities of the substrates but also fresh insights into the suitability of various models for describing the relationships between the kinetic and thermodynamic physical quantities.

## 1. Introduction

Over the past two decades, boron chemistry has reemerged as one of the most interesting fields in organic and inorganic chemistry. These new developments, which differ from traditional boron chemistry research fields [1], mainly focus on two aspects, namely, boron–boron multiple bonds [2,3] and the chemistry of coordinated borane compounds [4,5,6,7], the latter of which is the focus of the present study.

Owing to its extreme electron deficiency, the triatomic five-valence-electron ^•^BH_2_ free radical, which has a ground state of ^2^A_1_ [8], is much less thermodynamically stable than free BH_3_. As a result, BH_3_ exhibits strongly endothermic B–H homolytic cleavage and a high B–H bond dissociation energy (BDE) of approximately 105 kcal mol^−1^ [9,10]. The strong endothermicity likely increases the B–H bond cleavage barrier [11,12], making B–H bond activation difficult. Therefore, free BH_3_ is not a good H-donor or ^•^BH_2_ source.

The electron-poor features of boron allow BH_3_ to form complexes with electron-pair donors such as (Me)_n_NH_3−n_ (n = 0, 1, 3) [9,10,13] and R_3_P (R = Ph, Bu) [14], which lowers the B–H BDE remarkably to 90–101 kcal mol^−1^ [11]. This type of complexation leads to rich boron chemistry [4]; in particular, the coordination of boranes with N-heterocyclic carbenes (NHCs) as strong electron-pair donors gives very stable trivalent NHC-supported borane compounds (NHC-boranes) [5,6]. These stable neutral NHC-boranes are easily prepared by Lewis base exchange with amine-, phosphine- and dimethylsulfid-boranes [4,5,15] or by the reactions of the in situ generated NHC with borane in THF [16,17,18]. The B–H homolytic cleavage of NHC-boranes provides highly thermodynamically stable NHC-boryl radicals owing to effective stabilization via single-electron delocalization in the π-conjugated systems of NHCs, which substantially decreases the energies of NHC-boryl radicals relative to those of NHC-boranes. Thus, the introduction of an NHC can markedly lower the B–H BDE to 80–88 kcal mol^−1^ [19,20].

Thorough investigations of this new NHC-borane family [5,6] have greatly extended main group boron chemistry beyond classical boron compounds. The B–H bond activated by an NHC can be used directly in various types of reactions, such as hydroborations of unsaturated bonds [21,22], halogenations with halogen-based electrophiles [23] and B–H bond insertion reactions [24]. Furthermore, NHC-boranes are good H donors, as their highly hydridic nature results in easy B–H bond cleavage accompanied by the generation of NHC-boryl radicals. NHC-boranes have been used as substrates, reagents and coinitiators in various types of organic reactions, including the radical borylative cyclization of *N*-allylcyanamides [25,26], the reduction of xanthates [20] and the hydroxymethylation of organoiodides [27]. In the typical examples of reductive decyanation of several organocyanides [17,18,28,29], the addition of the NHC-^•^BH_2_ radical to R-CN provides NHC-BH_2_C(R)=N^•^ as an adduct intermediate. The subsequent elimination of R^•^ gives the NHC-boryl monocyanide NHC-BH_2_CN. The ability of the radical R^•^ to abstract H from NHC-BH_3_ and NHC-BH_2_CN determines not only the yield of NHC-BH_2_CN but also that of the NHC-boryl dicyanide NHC-BH(CN)_2_ [25,28]. Furthermore, the formation of NHC-^•^BH_2_ can act as a controlling factor in some reactions by effectively initializing a catalytic cycle [5]. Clearly, in those organic transformations with radical mechanism, hydrogen atom abstraction (HAA) from NHC-boranes is a very important step with the NHC-borane acting as a H source to terminate other radicals, as a stoichiometric reagent/substrate to provide boryl radicals for further reactions or as a catalyst [5]. Thus, it is exceedingly important to evaluate and quantitate the HAA reactivities of NHC-boranes for both scientific and practical purposes, such as understanding the factors that affect the kinetic barriers and their compositions, selecting suitable substrates and predicting the mechanisms and products of relevant reactions. 

It should be noted that the thermodynamic quantity BDE is not a true parameter reflecting the HAA reactivities of different NHC-boranes but instead a kinetic barrier. For an NHC-borane, the B-site substituent can affect the steric environment of the B center and the barrier to its HAA reaction. Furthermore, it should be possible to use the polar effect [30,31,32] to estimate the HAA reactivities of NHC-boranes. Substituting the B-site H of an NHC-borane with an electron-withdrawing group weakens the nucleophilicities of the H atom being abstracted and the B atom of the resulting NHC-boryl radical, which theoretically leads to weaker polarity matching with the electrophilic abstracting radical and a higher HAA barrier. Thus, electron-withdrawing groups on B likely have a detrimental effect on the HAA reactions of NHC-boranes, which may be the predominant reason for B-site disubstituted NHC-boranes being obtained in poor yields [17,25,28,29]. However, such conjecture requires further confirmation.

The improved Marcus equation [11,12] can be written as:(1)$$\Delta E^{\neq }=\Delta E_{0}^{\neq }+\frac{1}{2}\Delta_{r}E+\frac{{\left(\Delta_{r}E\right)}^{2}}{16\Delta E_{0}^{\neq }}.$$

This equation displays that the activation barrier ($\Delta E^{\neq }$) of a reaction consists of the intrinsic barrier ($\Delta E_{0}^{\neq }$) and the thermal contribution of the thermodynamic reaction energy to the kinetic barrier ($\Delta E_{\mathrm{therm}}^{\neq }$, the sum of the last two terms in Equation (1), hereafter referred to as the thermal contribution). Note, as $\Delta E^{\neq }$ includes the activation zero-point vibrational energy (ZPVE) correction ($\Delta E_{\mathrm{ZPVE}}^{\neq }$), it is a vibrationally adiabatic barrier [33]. The free energy barrier ($\Delta G^{\neq }$) can be readily obtained by introducing the temperature-dependent activation Gibbs free energy thermal correction ($\Delta G_{\mathrm{corr}}^{\neq }$) and the activation ZPVE correction into the activation barrier, as follows:(2)$$\Delta G^{\neq }=\Delta E^{\neq }+\Delta G_{\mathrm{corr}}^{\neq }-\Delta E_{\mathrm{ZPVE}}^{\neq }$$

Or
(3)$$\Delta G^{\neq }=\Delta E_{0}^{\neq }+\Delta E_{\mathrm{therm}}^{\neq }+\Delta G_{\mathrm{corr}}^{\neq }-\Delta E_{\mathrm{ZPVE}}^{\neq }.$$

Barriers are the basic physical quantities used to calculate reaction rate constants and to estimate reactivities. The intrinsic barrier can be considered a pure kinetic effect, while the activation and free energy barriers involve thermodynamic energy changes between products and reactants in addition to kinetic effects. Clearly, both the thermal contribution and the activation Gibbs free energy thermal correction in Equations (2) and (3) complicate the correlation of the temperature-dependent free energy barrier with the temperature-independent activation and intrinsic barriers. The intrinsic barrier and the thermal contribution are difficult to obtain directly from experiments but it is feasible to use the experimental activation barrier, which is mainly related to the activation energy (E_a_, Eyring barrier) and the thermodynamic reaction energy. However, the separation of the intrinsic barrier and the thermal contribution from the activation or free energy barriers is readily achieved using theoretical chemistry methods [11,12]. For the HAA reactions of NHC-boranes, such an approach can aid in understanding the compositions of barriers and the effect of the kinetic and thermodynamic components on their HAA reactivities.

In the field of physical organic chemistry, the correlation between the free energy barrier and the reaction Gibbs free energy remains of particular interest because the former is the key kinetic physical quantity for determining the reaction rate constant and the latter is the thermodynamic driving force of the reaction. Many types of reactions show good linear correlations, that is, following the Evans–Polanyi linear free energy relations [34,35]. However, Mayer’s pioneering works [36,37,38] indicated that for HAA reactions, the simplest class of proton-coupled electron transfer (PCET) process [39,40], the intrinsic barrier is an important variable in the functional relationship between the free energy barrier and the reaction Gibbs free energy, which shows a nonlinear correlation that can be well expressed by the three-variable Marcus-theory-based HAA model [37]. Under low driving force conditions, the Marcus-theory-based HAA model can be simplified to the Evans–Polanyi correlation model [37]. However, for the HAA reactions of NHC-boranes, the most suitable model for the relationship between the free energy barrier and the reaction Gibbs free energy remains unknown. 

To obtain a clearer understanding of the above-mentioned issues, in this investigation, we selected a series of NHC-boranes in the B-site is mono- or disubstituted with –F, –CN, and/or –Ph as substrates. Using ^•^CH_2_CN, Me^•^ and Et^•^ radicals as abstractors, the HAA reactions of these NHC-boranes were examined with a focus on the following aspects. First, the components of the barriers and the changes caused by varying the B-site substituents were explored. Second, the polar effect of the NHC-boranes and the abstracting radicals on the HAA reactivities of NHC-boranes were assessed. Third, the effect of the B-site substituents of the NHC-boranes on their B–H BDEs was studied to develop a method for predicting the B–H BDEs using the HAA reactions. Finally, a three-variable linear model (see: Appendix A) was suggested to describe the relationship between the kinetic and thermodynamic quantities for the HAA reactions of NHC-boranes. This model was shown to be more intuitive than the Marcus-theory-based nonlinear HAA model [37] and more accurate than the two-variable Evans–Polanyi linear model [34,35]. We hope that the present work will aid in improving experimental design and provide a deeper insight into physical-organic models with the aim of obtaining a better understanding of the physical quantities that affect reactivity.

## 2. Results and Discussion

The investigated HAA reactions are shown in Scheme 1. The selected NHC-boranes, which act as H donors, include frequently used diMe-Imd-BH_3_ (**1**) [5,16,19,21,25,28], those with –F, –CN, and/or –Ph substituents [5,6,25,28,29] and two special NHC-ligated boryl monocyanides, namely, tetraMe-Imd-BH_2_CN (**6**) and diiPr-Imd-BH_2_CN (**8**) [28]. The selected attacking radicals (abstractors) are methyl (Me^•^), ethyl (Et^•^) and cyanomethyl (^•^CH_2_CN). The relevant energies of the stationary points located at the B3LYP/6-311++G(d,p) and M06-2X/6-311++G(d,p) levels of theory are provided in Appendix A. In the following discussion, the temperature-dependent physical quantities were computed at 376 K, the boiling point of the solvent benzotrifluoride used in available experiments [23,28]. Furthermore, the energies in the fitted equations are in kcal mol^−1^.

### 2.1. Activation and Intrinsic Barriers and Thermal Contribution

Figure 1 and Appendix A display the percentages of the intrinsic barrier and the thermal contribution in the activation barrier for each of the investigated HAA reactions. For the HAA reactions with ^•^CH_2_CN, the percentages of the intrinsic barriers and the thermal contributions in the corresponding activation barriers vary by less than 9%, decreasing and increasing, respectively, with the increasing activation barrier. However, for the HAA reactions with Me^•^ and Et^•^, regardless of the activation barrier, the changes in the intrinsic barrier and thermal contribution are less than 2%.

The relative thermodynamic stabilities of the abstractors are predicted to be Me^•^ < Et^•^ < ^•^CH_2_CN based on the spin population at the central C atom (1.15, 1.10 and 0.87 e, respectively). The observed trend results from stabilization by $p\to \sigma_{C-H}^{*}$ hyperconjugation and stronger p→π conjugation in Et^•^ and ^•^CH_2_CN, respectively. Considering the small differences in the relative thermodynamic stabilities of the products and the closed-shell reactant for the HAA reactions of an NHC-borane with each of the three attacking radicals, we reasonably predict that the HAA reactions of an NHC-borane with Me^•^ and ^•^CH_2_CN should be more and less exothermic, respectively, than that with Et^•^. This deduction is supported by the computed results, as shown in Figure 2c and Appendix A. This stability sequence directly leads to a decreasing thermal contribution from Me^•^ to Et^•^ and then to ^•^CH_2_CN (−11.5 to −9.9, −9.9 to −8.3 and −8.0 to −6.4 kcal mol^−1^, respectively) (Figure 2b). The M06-2X-computed data gave similar results (−10.7 to −9.7, −9.1 to −8.2 and −7.5 to −6.4 kcal mol^−1^, respectively). However, the situation becomes somewhat complex when comparing the HAA reactions of a series of NHC-boranes with a given attacking radical.

The computed results indicate that the NHC-boryl radicals resulting from B–H cleavage are significantly stabilized by single-electron spin delocalization, as spin population of only 0.46–0.69 e are observed at the central B atoms (Appendix A). Because of the small changes in spin population (0.23 e), reaction energy (4.1 kcal mol^−1^) and thermal contribution (1.6 kcal mol^−1^) (Appendix A) for the three reaction series, the linear dependences of the reaction energies and thermal contributions on the spin population of the NHC-boryl radicals are rather weak without an acceptable goodness of fit (GOF) (Appendix A). Therefore, for the HAA reactions of a series of NHC-boranes with a given radical, the relationship of the reaction energy and the thermal contribution with the spin population of the NHC-boryl radicals cannot be determined. This result also implies that the changes in the reaction energy and the thermal contribution depend on not only the relative stabilities of the product NHC-boryl radicals but also those of the reactant NHC-boranes.

As shown in Figure 2f and Appendix A, the intrinsic barriers to the HAA reactions with Me^•^ and Et^•^ vary in the range of 16.6–19.8 kcal mol^−1^ (B3LYP) and 15.1–19.0 kcal mol^−1^ (M06-2X) and are somewhat higher than the corresponding values of 9.0–15.6 kcal mol^−1^ (B3LYP) and 7.5–13.3 kcal mol^−1^ (M06-2X) for the HAA reactions with ^•^CH_2_CN. The thermal contributions for the HAA reaction series with Me^•^ and Et^•^ are more favorable than those for the HAA reactions with ^•^CH_2_CN (Figure 2b). However, the activation barriers (Figure 2e) are not rearranged according to the thermal contribution ordering instead of the intrinsic barrier ordering, although the barrier difference is decreased. For example, the activation barriers for the HAA reactions of NHC-boranes with Me^•^ and Et^•^ vary in the range of 6.4–10.1 kcal mol^−1^ (B3LYP) and 6.7–9.2 kcal mol^−1^ (M06-2X), while the corresponding values for the HAA reactions with ^•^CH_2_CN are 2.6–7.8 kcal mol^−1^ (B3LYP) and 1.0–6.0 kcal mol^−1^ (M06-2X) (Figure 2e and Appendix A). Clearly, the relative activation barriers for the HAA reactions with Me^•^, Et^•^ and ^•^CH_2_CN are determined by the intrinsic barrier rather than the thermal contribution.

To understand the dependence of the activation barrier on the intrinsic barrier and the thermal contribution, we performed a correlation analysis. The fitting (Figure 3a) between activation and intrinsic barriers has an acceptable GOF with CODs of 0.88 and 0.94 for the HAA reactions with Me^•^ and Et^•^, respectively; slightly low CODs of 0.80 are obtained for both reaction series at the B3LYP level (Figure 3b). For the HAA reactions with ^•^CH_2_CN, the activation barrier is extremely dependent on the intrinsic barrier, with CODs of 0.98 and 0.95 at the M06-2X and B3LYP levels, respectively. Furthermore, the observed positive correlations indicate that the activation barrier increases with the increasing intrinsic barrier. However, the degree of correlation between the activation barrier and the thermal contribution was very low for each of the three HAA reaction series, as indicated by CODs lower than 0.36 (B3LYP) and 0.54 (M06-2X) (Figure 3). These results are, to some extent, related to the relatively large and small change ranges in the intrinsic barriers (2.8–6.6 kcal mol^−1^) and the thermal contributions (1.7–1.8 kcal mol^−1^), respectively (Appendix A).

Although the activation barriers for the three series of HAA reactions show acceptable linear correlations with the intrinsic barriers, the introduction of the thermodynamic reaction energies could improve the correlation because the intrinsic barrier is separated from the activation barrier by the thermodynamic reaction energy. As shown in Equation (1), the activation barrier is nonlinearly related to the thermodynamic reaction energy and thus likely decreases the linear correlation between the activation barrier and the intrinsic barrier. To estimate the effect of the reaction energy, we directly introduced this parameter into the fitting model (more mathematical physical treatment, see: Appendix A). The fitted equations obtained using the three-variable linear model and the B3LYP-computed data are as follows:(4)$$\Delta E^{\neq }=0.87\Delta E_{0}^{\neq }+0.32\Delta_{r}E+0.01\;(\mathrm{for}\;{\mathrm{Me}}^{•})$$
(5)$$\Delta E^{\neq }=0.91\Delta E_{0}^{\neq }+0.36\Delta_{r}E+0.04\;(\mathrm{for}\;{\mathrm{Et}}^{•})$$
(6)$$\Delta E^{\neq }=0.86\Delta E_{0}^{\neq }+0.32\Delta_{r}E+0.04\;(\mathrm{for}{\;}^{•}{\mathrm{CH}}_{2}\mathrm{CN})$$

At the M06-2X level, the corresponding equations are as follows:(7)$$\Delta E^{\neq }=0.88\Delta E_{0}^{\neq }+0.33\Delta_{r}E+0.05\;(\mathrm{for}\;{\mathrm{Me}}^{•})$$
(8)$$\Delta E^{\neq }=0.90\Delta E_{0}^{\neq }+0.34\Delta_{r}E-0.05\;(\mathrm{for}\;{\mathrm{Et}}^{•})$$
(9)$$\Delta E^{\neq }=0.80\Delta E_{0}^{\neq }+0.24\Delta_{r}E-0.50\;(\mathrm{for}{\;}^{•}{\mathrm{CH}}_{2}\mathrm{CN})$$

Surprisingly, the fitting using the three-variable linear model has a very high GOF with a COD of 1.0 for each reaction series at the B3LYP and M06-2X levels. The computational data expressed by the coordinates $\left(\Delta E^{\neq },\Delta E_{0}^{\neq },\Delta_{r}E\right)$ (in red) are located on a perfect plane (in blue), as shown in Figure 4 (B3LYP) and Appendix A (M06-2X). Furthermore, the linear fits of the projection points (in olive) on the $\left(\Delta E^{\neq },\Delta E_{0}^{\neq }\right)$ coordinate plane have CODs of 0.88, 0.94 and 0.95 (M06-2X) and 0.80, 0.80 and 0.98 (B3LYP) for the HAA reaction series with Me^•^, Et^•^ and ^•^CH_2_CN, respectively, as mentioned above. However, the linear fits of the projection points (in violet) on the $\left(\Delta E^{\neq },\Delta_{r}E\right)$ coordinate plane have a very low GOF with CODs of only 0.23 (Me^•^), 0.18 (Et^•^) and 0.02 (^•^CH_2_CN) at the B3LYP level and 0.17 (Me^•^), 0.23 (Et^•^) and 0.19 (^•^CH_2_CN) at the M06-2X level. In addition, for each of the three fitted plane equations, the coefficient of the dimension $\Delta E_{0}^{\neq }$ is larger than that of the dimension $\Delta_{r}E$, which means that the fitted plane has a larger dihedral angle with the $\left(\Delta E^{\neq },\Delta E_{0}^{\neq }\right)$ coordinate plane than with the $\left(\Delta E^{\neq },\Delta_{r}E\right)$ coordinate plane. This observation can be readily confirmed by the calculated results, as the two-dimensional surfaces expressed by the three fitted equations have dihedral angles of approximately 76° and 50° with the $\left(\Delta E^{\neq },\Delta E_{0}^{\neq }\right)$ and $\left(\Delta E^{\neq },\Delta_{r}E\right)$ coordinate planes, respectively. Clearly, although $\Delta E^{\neq }$ is highly linearly correlated with $\Delta E_{0}^{\neq }$, the smaller dihedral angle of the fitted planes with the $\left(\Delta E^{\neq },\Delta_{r}E\right)$ coordinate plane enlarges the dispersion of the projection points on the $\left(\Delta E^{\neq },\Delta_{r}E\right)$ coordinate plane, which leads to a high fitting deviation and low correlation between $\Delta E^{\neq }$ and $\Delta_{r}E$, as shown in Figure 4 and Appendix A.

### 2.2. Free Energy Barrier

Appendix A display the contributions and percentages of the intrinsic barrier, thermal contribution, activation ZPVE correction and activation Gibbs thermal correction to the free energy barrier. Because the activation ZPVE corrections for the HAA reaction series with Me^•^, Et^•^ and ^•^CH_2_CN vary by only 1.0, 0.4 and 0.6 kcal mol^−1^, respectively and the percentages of the Gibbs free energy are less than 2%, the effect of the activation ZPVE correction is ignored in the subsequent discussion.

For the HAA reactions of NHC-boranes with Me^•^ and Et^•^, the percentages of the intrinsic barrier and the activation Gibbs thermal correction in the free energy barrier are higher and lower than 39%, respectively (Appendix A), indicating that the free energy barrier is dominated by the intrinsic barrier. However, for the HAA reactions of NHC-boranes with ^•^CH_2_CN, the activation Gibbs thermal correction is predominant relative to the intrinsic barrier (38–51% and 26–40%, respectively, at the M06-2X level; 34–43% for both at the B3LYP level). Furthermore, the percentages of the thermal contribution in the free energy barrier are in the ranges of 25–29% (Me^•^), 21–24% (Et^•^) and 20–24% (^•^CH_2_CN) at the B3LYP level and 24–26% (Me^•^), 21–23% (Et^•^) and 21–25% (^•^CH_2_CN) at the M06-2X level. Although the thermal contributions for the three reaction series are somewhat different, the variation within each reaction series is very similar, approximately 1.6 kcal mol^−1^ (B3LYP, Appendix A and Appendix A) and 1.1 kcal mol^−1^ (M06-2X, Appendix A and Appendix A).

When the intrinsic barrier and thermal contribution are combined as the activation barrier, among the three components of the free energy barrier, the activation barrier is smaller in proportion than the activation Gibbs thermal correction; the latter has ranges of 60–80%, 56–64% and 53–62% (B3LYP, Appendix A) and 67–92%, 54–65% and 58–68% (M06-2X, Appendix A) for the ^•^CH_2_CN, Me^•^ and Et^•^ reaction series, respectively. Therefore, the thermal effect leads to a significant increase in barrier height from the activation barrier to the free energy barrier (Appendix A). Furthermore, for the ^•^CH_2_CN reaction series, the proportion of the activation Gibbs thermal correction decreases with the increasing free energy barrier, whereas the opposite trend is observed for the other two reaction series (Appendix A).

As shown in Figure 2d, the F-, Ph-, and/or CN-substituted NHC-boranes possess higher free energy barriers than unsubstituted NHC-BH_3_. Although the order of the free energy barriers for the HAA reaction series with Et^•^ and Me^•^ are very similar, that for the reaction series with ^•^CH_2_CN differs significantly. Furthermore, for the HAA reactions with ^•^CH_2_CN, CN-substituted NHC-boranes **5**‒**10** have higher free energy barriers than F- and/or Ph-substituted NHC-boranes **2**‒**4**. However, for the HAA reactions of NHC-boranes with Et^•^ and Me^•^, mononitrile-substituted **5** has a much lower HAA free energy barrier than the other substituted NHC-boranes.

Figure 5 displays the linear dependence of the free energy barrier on the intrinsic barrier, activation barrier, thermal contribution and activation Gibbs thermal correction. For the HAA reactions with ^•^CH_2_CN, the free energy barrier shows high linear correlation with the intrinsic and activation barriers with CODs of 0.93 and 0.85, respectively, at the B3LYP level, which decrease to 0.63 and 0.65, respectively, at the M06-2X level (Appendix A). However, the free energy barrier shows low correlation with the thermal contribution and the activation Gibbs thermal correction, as indicated by their CODs of 0.24 and 0.44, respectively, at the B3LYP level (Figure 5) and 0.33 and 0.10, respectively, at the M06-2X level (Appendix A). For the HAA reactions with Me^•^ and Et^•^, the free energy barriers are highly correlated with the activation Gibbs thermal correction with CODs of 0.95 (Me^•^) and 0.87 (Et^•^) at the B3LYP level (Figure 5), although the CODs at the M06-2X level are somewhat lower (0.82 for Me^•^ and 0.40 for Et^•^) (Appendix A). However, the free energy barrier is lower than the intrinsic and activation barriers and the thermal contribution, as shown in Figure 5 and Appendix A.

These results are closely related to the observed variations in these quantities. The HAA reactions with ^•^CH_2_CN exhibit larger changes in the intrinsic barrier (6.6 kcal mol^−1^) and the activation barrier (5.2 kcal mol^−1^) than in the thermal contribution (1.6 kcal mol^−1^) and the activation Gibbs thermal correction (2.3 kcal mol^−1^) (Appendix A). Thus, the former two parameters should show a greater dependency on the free energy barrier. Similarly, the changes in the activation Gibbs thermal correction for the HAA reactions with Me^•^ (6.5 kcal mol^−1^) and Et^•^ (4.6 kcal mol^−1^) are larger than those (<3.2 kcal mol^−1^) in the intrinsic and activation barriers and the thermal contribution. This behavior should be responsible for the free energy barrier showing greater correlation with the activation Gibbs thermal correction than with the other three parameters.

Another noteworthy finding is that for the three HAA reaction series, the free energy barrier shows low linear correlation with the reaction Gibbs free energy, with CODs smaller than 0.2 (Appendix A) and thus does not satisfy the Evans–Polanyi linear correlation model [34,35]. This deviation can be attributed to the high thermodynamic driving force (Appendix A), as the Marcus-theory-based HAA model can only be simplified to the Evans–Polanyi correlation under low driving force conditions [37]. To obtain deeper insights into the effect of the intrinsic barrier on the correlation between the free energy barrier and the reaction Gibbs free energy, we introduced the intrinsic barrier as a variable in the correlation analyses. The three-variable linear equations fitted at the B3LYP level are as follows:(10)$$\Delta G^{\neq }=2.23\Delta E_{0}^{\neq }+1.17\Delta_{r}G+6.63\;(\mathrm{for}\;{\mathrm{Me}}^{•})$$
(11)$$\Delta G^{\neq }=1.62\Delta E_{0}^{\neq }+0.81\Delta_{r}G+9.34\;(\mathrm{for}\;{\mathrm{Et}}^{•})$$
(12)$$\Delta G^{\neq }=1.04\Delta E_{0}^{\neq }+0.60\Delta_{r}G+15.45\;(\mathrm{for}{\;}^{•}{\mathrm{CH}}_{2}\mathrm{CN})$$

These fittings have a high GOF with CODs of 0.94, 0.93 and 0.97, respectively. At the M06-2X level, the CODs are 0.85, 0.88 and 0.86, respectively (Appendix A). Furthermore because the free energy barrier and the activation barrier are poorly correlated, as described above, we also substituted the activation barrier for the intrinsic barrier to explore the roles of the activation barrier and reaction Gibbs free energy in the correlations between the free energy barrier and the reaction Gibbs free energy and between the free energy barrier and activation barrier, respectively. Fitting using the B3LYP-computed data gave the following three-variable linear equations:(13)$$\Delta G^{\neq }=3.32\Delta E_{}^{\neq }+0.59\Delta_{r}G+8.63\;(\mathrm{for}\;{\mathrm{Me}}^{•})$$
(14)$$\Delta G^{\neq }=1.84\Delta E_{}^{\neq }+0.33\Delta_{r}G+12.46\;(\mathrm{for}\;{\mathrm{Et}}^{•})$$
(15)$$\Delta G^{\neq }=1.27\Delta E_{}^{\neq }+0.30\Delta_{r}G+16.72\;(\mathrm{for}{\;}^{•}{\mathrm{CH}}_{2}\mathrm{CN})$$
which had CODs of 0.92, 0.94 and 0.97, respectively, indicating a high linear GOF. The results fitted at the M06-2X level are shown in Appendix A and the corresponding CODs are in the range of 0.83–0.88. As a representative example, the planes (in blue) fitted using the three-variable linear model and the B3LYP-computed data points (in red) for the reactions with ^•^CH_2_CN are shown in Figure 6 (the results at the M06-2X level are shown in Appendix A), in which the projections points (in olive) on the $\left(\Delta G^{\neq },\Delta_{r}G\right)$ coordinate plane give a low linear GOF with a COD of only 0.06, whereas the projections points (in violet) on the $\left(\Delta G^{\neq },\Delta E_{0}^{\neq }\right)$ (Figure 6a) and $\left(\Delta G^{\neq },\Delta E_{}^{\neq }\right)$ (Figure 6b) coordinate planes indicate strong linear correlation of the free energy barrier with the intrinsic and activation barriers (CODs of 0.85 and 0.93, respectively).

The two sets of fitted equations indicate that the present reaction systems do not satisfy the Evans–Polanyi linear correlation model. Evidently, the introduction of the activation and intrinsic barriers improves the correlations of the free energy barrier with the reaction Gibbs free energy; on the other hand, the introduction of the reaction Gibbs free energy improves the correlations of the free energy barrier with the intrinsic and activation barriers. This simple three-variable linear model for describing the relationship between the kinetic barriers and the thermodynamic reaction Gibbs free energy seems not only more intuitive than the somewhat complex nonlinear Marcus-theory-based HAA model but also more rigorous than the two-variable Evans–Polanyi linear model. Thus, this simple three-variable linear model is suitable for the present HAA reactions of NHC-boranes.

### 2.3. Understanding Barriers and Reactivities Based on the Polarity Effect

Generally, difficulties in atom-abstraction reactions from a closed-shell molecule by a radical are directly related to the nucleophilic and electrophilic characteristics of the reactant and product radicals [30,31,32]. For radicals R^1^ and R^2^ in the HAA reaction R^1^H + R^2^→[R^1^∙∙∙H∙∙∙R^2^]^≠^→R^1^ + R^2^H, the stronger the nucleophilicity of one radical and the electrophilicity of the other, the better the polarity matching, which results in a lower kinetic barrier. Thus, this reaction can be understood as a typical electron push–pull effect in PCET [39,40].

For the HAA reactions investigated in this study, the orbital analyses of the located transition states indicated that radical electrophilic attack (electron transfer: $\sigma_{B-H}^{}\to \mathrm{SOMO}$) has an orbital interaction energy that is approximately ten times higher than that of radical nucleophilic attack (electron transfer $\sigma_{B-H}^{*}\leftarrow \mathrm{SOMO}$) (Appendix A). A representative example is shown in Figure 7. These results clearly suggest that electrophilic attack by the reactant radical is dominant in the HAA reactions of NHC-boranes.

In the following discussion, we first consider how the electrophilicities of the attacking radicals ^•^CH_2_CN, Me^•^ and Et^•^ affect the reaction kinetics. The calculated results give global electrophilicity indices of 34.90, 13.97 and 11.31 eV and local electrophilicity indices of 30.37, 16.06 and 12.44 eV for ^•^CH_2_CN, Me^•^ and Et^•^, respectively. Clearly, the strongly electron-withdrawing –CN group in ^•^CH_2_CN and the weakly electron-donating –CH_3_ group in Et^•^ are responsible for these trends. Furthermore, the order of the electrophilicity indices agrees well with that of the intrinsic, activation and free energy barriers for the HAA reactions of unsubstituted and F- and/or Ph-substituted NHC-boranes **1**–**4** (Figure 2). Thus, the relative reactivities can be determined directly using the electrophilicity indices of the attacking radicals. However, when a –CN group is introduced (NHC-boranes **5**–**10**), the situation becomes somewhat complex. As shown in Figure 2e, when the H atom is abstracted from NHC-borane **5**–**9** by ^•^CH_2_CN, Me^•^ and Et^•^, the order of the activation barriers is Et^•^ > Me^•^ > ^•^CH_2_CN, which is consistent with that of their electrophilicity indices. However, the intrinsic barriers for the HAA reactions of **5**–**10** with Me^•^ and Et^•^ are similar to and higher than, respectively, those with ^•^CH_2_CN, whereas the free energy barriers for the HAA reactions with ^•^CH_2_CN and Me^•^ are similar to and lower than, respectively, those with Et^•^. These results indicate that the intrinsic, activation and free energy barriers for the HAA reactions of a given NHC-borane with different radicals can only be estimated using the electrophilicity indices of the attacking radicals if these indices are sufficiently different.

Second, we assessed the barrier changes for the substituted NHC-boranes relative to that of NHC-BH_3_ (**1**) when the B-site H atom is abstracted by a given radical. Substitution with electron-withdrawing –CN, –Ph, and/or –F groups weakens the electron-rich characteristics of the B–H bond and the boron center in the corresponding NHC-boryl radical, thus decreasing their nucleophilic abilities. As a result, the polarity matching of the electrophilic attacking radical with the substituted NHC-boranes is weaker than that with NHC-BH_3_. Thus, the HAA barrier is expected to increase when –CN, –Ph, and/or –F groups are introduced into NHC-BH_3_. This behavior is well supported by the calculated results, as the free energy barriers for the HAA reactions of NHC-BH_3_ with ^•^CH_2_CN, Me^•^ and Et^•^ (13.8, 14.7 and 18.9 kcal mol^−1^, respectively) increase to 16.0–21.0, 17.4–20.6 and 21.0–23.8 kcal mol^−1^, respectively, after substitution (Appendix A). These results are also in good accordance with available experimental data [17,25,28,29]. For instance, addition–elimination reactions between NHC-boryl radicals and cyano compounds give NHC-boryl monocyanides but further H-abstraction–addition–elimination reactions to provide NHC-boryl dicyanides are generally difficult. Apparently, increasing the barrier height prevents further H-abstraction–addition–elimination reactions for NHC-boryl monocyanides.

Third, we evaluated the relative reactivities of the substituted NHC-boranes as H donors. When using ^•^CH_2_CN as the attacking radical, the global electrophilicity and nucleophilicity indices of the closed-shell NHC-boranes show moderate linear correlations with the intrinsic, activation and free energy barriers (CODs of 0.52–0.86 (B3LYP) and 0.31–0.94 (M06-2X)). These correlations are slightly stronger than the linear dependencies of the three barriers on the charges of the borane fragments or the H atoms abstracted by ^•^CH_2_CN (CODs of 0.21–0.68 (B3LYP) and 0.10–0.59 (M06-2X)) but slightly weaker than the linear correlations with the global and local electrophilicity/nucleophilicity indices of the open-shell NHC-boryl radicals (CODs of 0.67–0.89 (B3LYP) and 0.54–0.84 (M06-2X)) (Appendix A). Relative to the reactions with ^•^CH_2_CN, the HAA reactions with Me^•^ and Et^•^ show a very low GOF with CODs of only 0.01–0.32 at the B3LYP and M06-2X levels (Appendix A). This result can be attributed to Me^•^ and Et^•^ having weaker electrophilicities than ^•^CH_2_CN but not to differences in the nucleophilicities of NHC-boranes or the corresponding NHC-boryl radicals. In other words, the small changes in the barriers (Appendix A) originating from the relatively weak electrophilicities of Me^•^ and Et^•^ lead to poor correlations between the barriers and the nucleophilicity/electrophilicity indices of NHC-boranes or the corresponding NHC-boryl radicals.

### 2.4. Bond Dissociation Energies

The computed B–H BDEs of the NHC-boranes are in the range of 73.1–77.2 kcal mol^−1^ (B3LYP) and 75.8–78.1 kcal mol^−1^ (M06-2X), whereas the bond dissociation enthalpies (BDHs) are only 1.5–1.9 kcal mol^−1^ (B3LYP) and 1.3–2.7 kcal mol^−1^ (M06-2X) higher than the corresponding BDEs, as shown in Figure 8a. These results indicate that σ_B–H_ electron delocalization into the NHC π-conjugated system plays an important role in decreasing the B–H BDEs of NHC-boranes relative to those of BH_3_ and amine- and phosphine-coordinated boranes [9,10]. Furthermore, it was previously reported stated that the B–H BDEs of **L**-BH_3_ (**L** = H_2_O, NH_3_, PH_3_, CO, OCH_2_, NCH and CNH) are highly linearly correlated with the spin population of the central B atom of **L**-^•^BH_2_ [9]. However, for the present systems, the correlations of the BDEs and BDHs with the spin population of the central B atom of the radicals are very poor, with CODs of less than 0.2 (Appendix A). Clearly, the electron dispersion effects of the B-site substituents complicate the relationship of the BDEs and BDHs with the spin population of the B-centered radical products. Thus, similar to various other HAA reactions [36], the spin population of the unpaired electron is not a good predictor of BDEs and BDHs for the present substituted NHC-boranes.

As shown in Figure 8a, the ordering is approximately the sum for the calculated BDEs and the BDHs. Furthermore, for the substituted diMe-Imd-BH_3_ compounds, the B–H BDEs and BDHs are lower than those of unsubstituted diMe-Imd-BH_3_ (**1**). However, tetraMe-Imd-BH_2_CN (**6**) and diiPr-Imd-CH_2_CN (**8**) have higher B–H BDEs and BDHs than diMe-Imd-BH_3_ (**1**) and diMe-Imd-BH_2_CN (**5**), which indicates that the electron-donating effects of the additional Me groups in tetraMe-Imd and *i*-Pr in diiPr-Imd increase the electron densities of the NHC rings, resulting in more electron-rich B–H bonds with higher BDEs and BDHs.

For a series of NHC-boranes, the BDEs are strictly linearly correlated with the reaction energies ($\Delta_{r}E$) of their HAA reactions with radical R^•^ (see Appendix A), which is well supported by the linear fitting between the BDEs and $\Delta_{r}E$ (CODs of 1.0 for all the reaction series), as shown in Figure 8b and Appendix A. Thus, this correlation provides a convenient method for determining the BDE of an NHC-borane using a series of designed HAA reactions.

Because the reaction energy is closely related to the separated thermal contribution, we investigated the linear fitting between the thermal contribution and the BDEs. The results displayed in Figure 8b indicate that the B–H BDEs of NHC-boranes have an acceptable correlation with the thermal contribution of their HAA reactions with ^•^CH_2_CN, Et^•^ and Me^•^ with CODs of 0.77, 0.98 and 0.99, respectively. Therefore, when using attacking radicals without π-conjugation effects, the thermal contribution is expected to be a better predictor of the BDE of an NHC-borane than the spin population at the B atom of the NHC-boryl radical product.

## 3. Computational Details

All computations were carried out using the Gaussian09 program package [41]. Stationary points were fully optimized using the B3LYP [42,43,44] and Truhlar’s M06-2X [45] functionals with the McLean–Chandler 6-311G basis set [46], which has polarization and diffusion functions on hydrogen and heavier second-row atoms (i.e., 6-311++G(d,p)). The two functionals were selected based on available reports [47,48,49,50,51,52,53,54,55] and our tests. The theoretical methods used often in the available computations of NHC-boranes, involving H-atom abstraction, addition, cyclization and complexation reactions, are B3LYP [47,48,49,50,51] and M06-2X [52,53,54,55]. The M06-2X functional, a high-nonlocality functional with double the amount of nonlocal exchange (2X), was recommended for use in main-group thermochemistry, thermodynamic kinetics, noncovalent interactions and so forth. It gave good results for heavy- and H-atom-abstraction barrier height calculations, as reviewed in Truhlar’s pioneering work [45]. Our tests for a series of H-abstraction reactions of NHC-boranes indicated that the B3LYP-computed quantities are highly linearly correlative to the M06-2X-computed quantities, as observed that the coefficients of determinations (COD, R-squared, R^2^) are larger than 0.92 and 0.98 for H-abstraction barriers and thermodynamic reaction energies, respectively (more tests, see Appendix A). Although the B3LYP and M06-2X functionals gave a slightly different ordering of barriers to the investigated H-atom abstractions of NHC-boranes, the results computed at the two levels of theory keep well consistent in predicting the statistical correlation degrees of different kinetic and/or thermodynamic physical quantities. Thus, unless otherwise specified, the data used in the following discussion were those computed at the B3LYP level. When the results are method-dependent or need to be emphasized or compared, they are discussed at the two computational levels.

Harmonic vibration frequency calculations were conducted at the same level of theory with the corresponding geometric optimization. These calculations were used to characterize the located stationary points as equilibrium structures with real and positive frequencies or as first-order saddle points (transition states) with only one imaginary vibration mode. Furthermore, the bulk solvent effect was considered in all geometry optimizations and frequency computations using the self-consistent reaction field method [56,57] with the integral equation formalism polarizable continuum model of Tomasi and co-workers [58,59,60]. Benzotrifluoride was employed as the solvent in analogy with the experimental medium [23,28]. The located transition states were examined by the intrinsic reaction coordinate method [60] to determine the minima that they can reach along the reaction coordinate.

To consider the polar effect on the HAA reactions, the electrophilicity/nucleophilicity indices of reactants and products were computed at the mPWPW91/6-311+G(d,p) level of theory [61]. The global electrophilicity index (ω) of a species was calculated as $\omega =\mu^{2}/2\eta$ [62], where μ and η are the global chemical potential [63,64] and the global chemical hardness, respectively [64]. These parameters can be expressed by the ionization potential (I) and the electron affinity (A) as −(I + A) and I − A, respectively [64,65]. For closed-shell species, the energies of the highest occupied molecular orbital (HOMO) and the lowest unoccupied molecular orbital (LUMO) were employed to compute I and A, that is, $I=-E_{\mathrm{HOMO}}$ and $A=-E_{\mathrm{LUMO}}$. In contrast, for open-shell radicals, these parameters were calculated using the highest-energy α- and β-spin states ($E_{\mathrm{HOMO}}^{\alpha }$ and $E_{\mathrm{LUMO}}^{\beta }$), that is, $I=-E_{\mathrm{HOMO}}^{\alpha }$ and $A=-E_{\mathrm{LUMO}}^{\beta }$. The global nucleophilic indices (N) of the closed- and open-shell species were calculated as $E_{\mathrm{HOMO}}-E_{\mathrm{HOMO}}^{\alpha }(F^{•})$ and $E_{\mathrm{HOMO}}^{\alpha }-E_{\mathrm{HOMO}}^{\alpha }(F^{•})$ [65], respectively, using $E_{\mathrm{HOMO}}^{\alpha }(F^{•})$, the HOMO energy of the α-spin states of F^•^, as a reference. Furthermore, for the open-shell species, the local electrophilicity (ω_R_) and nucleophilicity (N_R_) indices were directly calculated by introducing the spin population (ρ_s_) of the radical center atom R, that is, $\omega_{R}=\omega \rho_{s}$ and $N_{R}=N\rho_{s}$ [66]. The spin population was determined from Mulliken population analysis at the mPWPW91/6-311+G(d,p) level of theory.

Furthermore, we conducted natural bond orbital (NBO) analyses for the located transition states and calculated their orbital interaction energies using the B3LYP/6-311++G(d,p)-computed transition state geometries. The NBO computations were carried out at the same level of theory using the NBO codes (Version 3.1) implemented in the Gaussian09 program package [41].

The Equation (1) can be rearranged as
(16)$$\Delta E_{0}^{\neq }=\frac{1}{2}\Delta E^{\neq }-\frac{1}{4}\Delta_{r}E+\frac{1}{2}\sqrt{{\left(\Delta_{r}E\right)}^{2}-\Delta_{r}E\Delta E_{}^{\neq }}.$$

Thus, the intrinsic barrier can be determined by the computed activation barrier and reaction energy. Further, the thermal contribution can be calculated using the equation
(17)$$\Delta E_{\mathrm{therm}}^{\neq }=\Delta E^{\neq }-\Delta E_{0}^{\neq }.$$

## 4. Conclusions

The HAA reactions of a series of NHC-boranes with ^•^CH_2_CN, Me^•^ and Et^•^ were examined in detail to obtain a better understanding of the relationship between kinetic and thermodynamic quantities. Based on the calculated data, the following conclusions can be drawn.

1. The HAA reactions of a given NHC-borane with Et^•^ and Me^•^ have similar intrinsic barriers. The thermal contribution for the HAA with Et^•^ is less favorable than that with Me^•^, which is the dominant factor in the former reaction have a higher activation barrier. Owing to ^•^CH_2_CN have a stronger electrophilicity than Et^•^ or Me^•^, HAA from a given NHC-borane by ^•^CH_2_CN has a significantly lower intrinsic barrier than the corresponding HAA by Et^•^ or Me^•^. Although the reaction energies and the thermal contributions for the HAA reactions with Et^•^ and Me^•^ are somewhat favorable, the high intrinsic barriers lead to higher activation and free energy barriers than for the HAA reaction with ^•^CH_2_CN.

2. The computational results for the orbital interaction energies of the HAA transition states verify that the HAA reactions of NHC-boranes involve electrophilic attacks by Me^•^, Et^•^ and ^•^CH_2_CN. The kinetic regulation of the HAA reactions is in accordance with the polarity matching rule, that is, strengthening the electrophilicity of the attacking radical or the nucleophilicity of the NHC-borane can effectively decrease the activation and free energy barriers for the HAA reaction. The polar effect can be used to estimate the relative reactivities of ^•^CH_2_CN, Et^•^ and Me^•^ for abstracting the H atoms of a given NHC-borane. The nucleophilicity and electrophilicity indices are not suitable for comparing the relative HAA reactivities of NHC-boranes with different B-site substituents because of the remarkably small differences in the barriers.

3. For NHC-boranes, the B–H BDEs cannot be used to predict their abilities as H donors. The coordination of an NHC significantly decreases the B–H BDE relative to that of free BH_3_, whereas the B-site substituents have only a minor effect on the B–H BDE. The HAA reaction energies of NHC-boranes are highly linearly correlated with the B–H BDEs and thus can be used to estimate the B–H BDE of a new NHC-borane. The thermal contribution can also be used for such predictions but the abstractors should be limited to radicals without π-conjugation effects.

4. The suggested three-variable linear model is suitable for describing the dependence of not only the activation barrier on the intrinsic barrier and the thermodynamic reaction energy but also the free energy barrier on the intrinsic (or activation) barrier and the reaction Gibbs free energy. This model has higher fitting and forecasting precision than the two-variable Evans–Polanyi linear model and is more transparent than the Marcus-theory-based nonlinear HAA model.

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
