# Peer review of "DFT Investigation of Hydrogen Atom Abstraction from NHC-Boranes by Methyl, Ethyl and Cyanomethyl Radicals—Composition and Correlation Analysis of Kinetic Barriers"

_molecules, 2020, doi:10.3390/molecules25194509_

Round 1
Reviewer 1 Report
The manuscript reports detailed computational studies on the hydrogen atom abstraction reactions of a series of ten NHC-boranes by three small radicals. The objective of the authors is to investigate energetic quantities that contribute to the reaction profile (kinetics and thermodynamics), to relate these quantities to chemical concepts as quantified, for example, by electrophilicity measures, and to establish simple parametric models that relate barriers with reaction energies. The three-variable model appears to be adequately predictive, although it is unclear to me whether it is a practically useful construct, given that the constituting quantities would have to be computed anyway. I cannot judge the novelty/originality of the work with respect to the specific field, but the study appears sound, provides ample details, and is presented in a clear way. I would only recommend to reduce the decimal digits to one when reporting energies in kcal/mol the text, and also to replace "spin density" or "spin distribution" with "spin population" when discussing atom-centered quantities derived from a spin population analysis. The type of spin population analysis should also be specified (Mulliken?). Finally, it was not clear to me why 376 K was chosen as the temperature.
Author Response
Dear Reviewer:
Thank you very much for your evaluation and comments on our manuscript. The comments and suggestions are extremely helpful to improve the quality of manuscript. We are very pleased to submit a revised manuscript, which have been technically corrected according to your and referee’s suggestions. In the attached separate document (Response to the first referee) included in the uploaded files, we have carefully addressed the issues raised by the reviewers, point by point. Furthermore, we also carefully checked the manuscript based on the referee’s suggestions. The comments and suggestions have significantly improved the quality of manuscript. We sincerely hope this revised manuscript can meet your standard.
Thank you very much for all your help.
Yours sincerely,
Prof. Hai-tao Yu (yuhaitao@hlju.edu.cn)
Prof. Hui Xu (hxu@hlju.edu.cn)
Sep. 15, 2020

Reviewer 2 Report
The manuscript “DFT Investigation of Hydrogen Atom Abstraction from NHC-boranes by Methyl, Ethyl, and Cyanomethyl Radicals: Composition and Correlation Analysis of Kinetic Barriers“ by Hong-jie Qu, Lang Yuan, Cai-xin Jia, Hai-tao Yu and Hui Xu represents a computational study of several hydrogen abstraction reactions of boron compounds. The paper is written in good English, and the calculations seem to be performed with due caution.
However, the scope of the study is too narrow, and the results are being over-interpreted. The authors introduce their work in the hope of providing useful information for the boron community, but the rest of the manuscript deals with too detailed analyses that do not provide considerable chemical insight and are hard to read even for a computational chemist. Overall, the manuscript is a routine study on a moderately interesting class of compounds and, as such, does not deserve in my opinion publication in the Molecules journal.
Author Response
Dear Reviewer:
Thank you very much for your evaluation and comments on our manuscript. The comments and suggestions are extremely helpful to improve the quality of manuscript. We are very pleased to submit a revised manuscript, which have been technically corrected according to your and referee’s suggestions. In the attached separate document (Response to the second referee) included in the uploaded files, we have carefully addressed the issues raised by the reviewers, point by point. Furthermore, we also carefully checked the manuscript based on the referee’s suggestions. The comments and suggestions have significantly improved the quality of manuscript. We sincerely hope this revised manuscript can meet your standard.
Thank you very much for all your help.
Yours sincerely,
Prof. Hai-tao Yu (yuhaitao@hlju.edu.cn)
Prof. Hui Xu (hxu@hlju.edu.cn)
Sep. 15, 2020

Reviewer 3 Report
The manuscript presents the computational study of the reaction barriers for a series of NHC-boranes with a methyl, ethyl and cyanomethyl radical reactions by using two DFT methods and solvent effects. The key findings presented in the manuscript are detailed composition and the correlations obtained for the reaction barriers in these hydrogen abstraction reactions. The manuscript is generally well written, with a promising abstract and well organized introduction. More computational details are necessary for calculations of intrinsic barriers and thermal contributions.
The reaction activation energies obtained are very low, which is not unusual for radical reactions. Tunneling effects and different possible reaction mechanisms (HAT, PCET, stepwise) were not taken into account. Unfortunately, in the literature there is no experimental data for comparison.
The excellent correlation obtained with three-variable model (Fig. 4) raises some suspicion for me. Although the thermal contribution is nonlinearly related to the thermodynamic reaction energy (Eq. 1), in a very narrow range of the obtained energies (few kcal mol-1) that could be the case. The authors should double check for the possibility of some lucky mutual compensation. Physical meanings of three-variable model and the obtained coefficients should also be discussed.
It is claimed that the proposed three-variable model is more appropriate than Evans–Polanyi or Marcus-theory-based models in this case, more evidence would be desirable for comparison.
Minor comments:
- Fig. 8a, is there any reason other than increasing values that the NHC-borane compounds are arranged in random order ?
- 598 - typo - Fruchtl.
Author Response
Dear Reviewer:
Thank you very much for your evaluation and comments on our manuscript. The comments and suggestions are extremely helpful to improve the quality of manuscript. We are very pleased to submit a revised manuscript, which have been technically corrected according to your and referee’s suggestions. In the attached separate document (Response to the third referee) included in the uploaded files, we have carefully addressed the issues raised by the reviewers, point by point. Furthermore, we also carefully checked the manuscript based on the referee’s suggestions. The comments and suggestions have significantly improved the quality of manuscript. We sincerely hope this revised manuscript can meet your standard.
Thank you very much for all your help.
Yours sincerely,
Prof. Hai-tao Yu (yuhaitao@hlju.edu.cn)
Prof. Hui Xu (hxu@hlju.edu.cn)
Sep. 15, 2020

Reviewer 4 Report
In this work, the authors studied the kinetics of hydrogen abstraction from boron complexes via three radicals (CH3, C2H5, CH3CN).
They have produced a large amount of data and reading the manuscript is a bit challenging. I would like to suggest simplifying the presentation.
However, all this information comes from quantum mechanical calculations done essentially at the B3LYP/6-311G level (a part also with the functional M06-2X). This is a very delicate point for the following reasons:
1) was the calculation "unrestricted"? It would seem so but it is not specified in the text.
2) The Kohn-Sham DFT has problems in the correct evaluation of the static correlation energy and in the inclusion of the dispersion energy (dynamic correlation contribution). In this study the transition state should have a certain multiconfigurational character
and therefore we must ask ourselves if the functional used is appropriate and if the results are in agreement with those obtainable from ab-initio multiconfigurational calculations with wave function. Differences with different functionals and different methods could be very significant both in absolute and relative values in the comparisons between different complexes.
3) The dispersion energy could stabilize the TS and could be different in the different cases treated.
Moreover (minor point): which PCM was used in the calculations? Reference 51 has been added in references to the PCM which is not relevant (page 4).
I suggest a substantial revision that addresses the points highlighted in this report.
Author Response
Dear Reviewer:
Thank you very much for your evaluation and comments on our manuscript. The comments and suggestions are extremely helpful to improve the quality of manuscript. We are very pleased to submit a revised manuscript, which have been technically corrected according to your and referee’s suggestions. In the attached separate document (Response to the fourth referee) included in the uploaded files, we have carefully addressed the issues raised by the reviewers, point by point. Furthermore, we also carefully checked the manuscript based on the referee’s suggestions. The comments and suggestions have significantly improved the quality of manuscript. We sincerely hope this revised manuscript can meet your standard.
Thank you very much for all your help.
Yours sincerely,
Prof. Hai-tao Yu (yuhaitao@hlju.edu.cn)
Prof. Hui Xu (hxu@hlju.edu.cn)
Sep. 15, 2020

Round 2
Reviewer 2 Report
I would like to thank the authors for their response. I agree with the authors that the boron chemistry by itself is a fascinating subject that deserves further attention. However, I do not think that proper insight is provided by the manuscript, it would be hard for other researchers to use the included information in further studies, and I expect very low interest for the readership. The authors write in their response that "The present study is a preliminary and exploratory investigation." For me, the amount of information given is too limited (although reproduced in many graphs) even for a preliminary study and therefore, it would need to be rewritten completely and provided with new data in order to be of interest for the community. As I do not think that routine studies with a narrow focus should be published in the literature (or at least not by this journal), I still recommend rejecting the manuscript.
Author Response
The explanation and change are included in the attached PDF document.

Reviewer 4 Report
The authors showed me some plots arising from further calculations in order to reply to my comments 2 and 3. The manuscript was not modified. I think that the authors should take the responsibility of their research strategy by adding something to the manuscript regarding the tests they followed to decide for the two functionals B3LYP and M06-2X. This should be done before publication.
Author Response

(The authors gave the same response as above.)
